# Anti-Arthritogenic Property of Interleukin 10-Expressing Human Amniotic MSCs Generated by Gene Editing in Collagen-Induced Arthritis

**DOI:** 10.3390/ijms23147913

**Published:** 2022-07-18

**Authors:** Dong-Sik Chae, Young-Jin Park, Sung-Whan Kim

**Affiliations:** 1Department of Orthopedic Surgery, International St. Mary’s Hospital, College of Medicine, Catholic Kwandong University, Incheon 22711, Korea; drchaeos@gmail.com; 2Department of Family Medicine, College of Medicine, Dong-A University, Dong-A University Medical Center, Busan 49201, Korea; yjpfm@dau.ac.kr; 3Department Medicine, College of Medicine, Catholic Kwandong University, Gangneung 25601, Korea

**Keywords:** cell therapy, collagen-induced arthritis, gene editing, mesenchymal stem cells, IL-10

## Abstract

Although stem cells are promising tools for the treatment of arthritis, their therapeutic effects remain controversial. In this study, we investigated the therapeutic properties of interleukin (IL)-10-overexpressing human amniotic mesenchymal stem cells (AMMs) generated via gene editing in a collagen-induced mouse model. IL-10 was inserted into the genomic loci of AMMs via transcription activator-like effector nucleases. In vitro immunomodulatory effects of IL-10-overexpressing AMMs (AMM/I) were evaluated and their anti-arthritogenic properties were determined in collagen-induced arthritis (CIA) mice. Transplantation of AMM/I attenuates CIA progression. In addition, the regulatory T cell population was increased, while T helper-17 cell activation was suppressed by AMM/I administration in CIA mice. Consistently, AMM/I injection increased proteoglycan expression, while reducing inflammation and the expression levels of the pro-inflammatory factors, IL-1 β, IL-6, monocyte chemoattractant protein-1, and tumor necrosis factor- α, in joint tissues. In conclusion, use of IL-10-edited human AMM/I may be a novel therapeutic strategy for the treatment of arthritis.

## 1. Introduction

Rheumatoid arthritis (RA) is systemic, chronic inflammatory and autoimmune disease that causes joint pain, swelling, lesions and articular cartilage destruction, eventually leading to the loss of joint function and disability [1]. RA can be a huge burden on health services. Various therapeutic interventions, such as anti-rheumatic drugs, tumor necrosis factor (TNF)-α inhibitors, and interleukin (IL)-1 and IL-6 blockades, have been used for RA [2,3]. However, these approaches face problems, such as high cost, side effects, and long treatment period [4]. Therefore, it is necessary to develop more effective and safe therapies for RA.

Pro-inflammatory cytokines play a key role in RA progression, and abnormal activation of T cells or macrophages contributes to the production of inflammatory cytokines [5,6]. IL-17 also induces the release of proinflammatory factors in synovial fibroblasts [7]. Stem cell therapy is used for the treatment of RA, and mesenchymal stem cells (MSCs) have the capacity to modulate immunity by releasing anti-inflammatory factors or indirectly inducing the polarization of macrophages or regulatory T cells (Tregs) [8,9]. MSCs are used for the treatment of RA of the knee [10]. MSCs show various therapeutic effects, including pain reduction, mild immune modulation, partial recovery of injured cartilage, and improved joint function without serious side effects [10]. However, more studies employing the latest technology are needed to increase their efficacy and safety.

IL-10 is a homodimeric cytokine that inhibits the production of proinflammatory cytokines, such as TNF-α and interferon-γ, in T cells or activated macrophages [11]. Suppression of IL-10 in synovial cells increases the levels of inflammatory cytokines, and systemic treatment with IL-10 suppresses the development of arthritis [12]. Additionally, IL-10-transduced murine MSCs inhibit experimental arthritis [13].

Recently, we developed a stable gene expression system at the genomic loci of human amniotic mesenchymal stem cells (AMMs) via transcription activator-like effector nucleases (TALENs) to facilitate long-term cytokine secretion [14]. In this study, we investigated the anti-arthritogenic properties of genome-edited IL-10-overexpressing human amniotic MSCs (AMM/I) in cartilage-damaged experimental arthritis.

## 2. Material & Methods

### 2.1. Cell Culture and Mouse

Human amniotic mesenchymal stem cells (AMMs) were purchased from Thermo Fisher Scientific, Inc. (Waltham, MA, USA). Database name/accession number of AMMs is Cytiva SV30103.01/10677394. The AMMs were cultured in low-glucose Dulbecco’s Modified Eagle Medium (DMEM; GIBCO, Grans Island, NY, USA) supplement with 10% Fetal bovine serum (FBS), 100 U/Ml penicillin and 100 µg/mL streptomycin (GIBCO) by previously reported [15]. Six-week old male DBA/1 mice were purchased from Orientbio (Seongnam, Korea).

### 2.2. Donor Vector Construction, Transfection and Selection

AMM/I was generated by previously described [15]. Briefly, IL-10 was synthesized and knock-in to the adeno-associated virus integration site 1 (AAVS1), targeting the donor vector (System Biosciences, Palo Alto, CA, USA) at the NdeⅠ and SalⅠ restriction sites (Appendix A). For electroporation, 1 × 10^5^ AMMs were suspended with 0.6 μg of AAVS1 left TALE-Nuclease vector, AAVS1 right TALE-Nuclease vector, and AAVS1 HR Donor (System Biosciences) in 10 μL electroporation buffer. The cells were electroporated using the Neon Transfection System (Thermo Fisher Scientific), and 5 days after transfection, IL-10 knock-in AMMs were selected by incubating with 5 μg/mL puromycin for 7 days. Puromycin-selected cells were resuspended in fluorescence-activated cell sorting (FACS) buffer and sored using FACS as previously described [15].

### 2.3. Genomic DNA Extraction and Junction PCR

Genomic DNA from the cultured cells was extracted using a G-spin™ Total DNA Extraction Mini Kit (Intron Biotechnology, Suwon, Korea) according to the manufacturer’s instructions. Next, 120 ng of genomic DNA was amplified by touch-down PCR (for 36 cycles) and second PCR as previously described [15].

### 2.4. Quantitative Reverse Transcription PCR (qRT-PCR)

qRT-PCR assays were performed as previously reported [16,17]. Briefly, total RNA was extracted from cells using RNA-stat (Iso-Tex Diagnostics, Friendswood, TX, USA) and isolated RNA was reverse-transcribed using TaqMan reagents (Applied Biosystems, Foster City, CA, USA). The synthesized cDNA was subjected to qRT-PCR using specific primers and probes. RNA levels were quantitatively examined by an ABI PRISM 7000 instrument (Applied Biosystems). Relative mRNA level was normalized to that of GAPDH expression. The qRT-PCR primers used were as follows: human IL-10 (Hs00961622_m1), and GAPDH (Hs99999905_m1), and mouse IL-1β (Mm00434228_m1), IL-6 (Mm00446190_m1), MCP-1 (Mm00441242_m1), TNF-α (Mm00443258_m1), and GAPDH (Mm99999915_g1). All primers and probe were purchased from Applied Biosystems.

### 2.5. Splenocyte Co-Culture and Enzyme-Linked Immunosorbent Assay (ELISA)

Splenocyte co-culture assay was conducted as previously reported [18]. Briefly, spleens from healthy DBA/1 mice were collected and minced in phosphate-buffered saline (PBS). Splenocytes were isolated using Ficoll-Hypaque density-gradient centrifugation and suspended in RPMI 1640 medium. To investigate the effects of AMM/I on T cells, 1 × 10^6^ AMMs or AMM/I were treated with or without 10 ng/mL TNF-α for 1 day and then co-cultured with 1 × 10^6^ splenocytes in RPMI 1640 containing 10% FBS. Supernatants from co-cultures were collected after 2 days, and cytokine levels were examined. The cytokine levels in the supernatant or serum were examined by murine IL-4 or IL-17A ELISA kits (R&D Systems, Minneapolis, MN, USA) according to the manufacturer’s specifications.

### 2.6. Induction of Collagen-Induced Arthritis Model and Treatment

Bovine type II collagen (Chondrex, Redmond, WA, USA) was emulsified at a ratio 1:1 with compelet Freund’s adjuvant (Chondrex) containing 2 mg/mL heat-killed *Mycobacterium tuberculosis*. 6-week-old male DBA/1 mice (n = 5 per group; OrientBio, seongnam, Korea) received a primary immunization, followed by a boosting immunization on day 21 using the same concentration of Bovine type II collagen and incomplete Freund’s adjuvant (Chondrex). Injection was intradermally conducted at the base of tail. The severity was observed for 28 days after first injection. The severity of arthritis was monitored and scored as determined by hind paw swelling and clinical scoring [19]. To evaluate therapeutic efficiency, 1 × 10^6^ of AMMs, AMM/I and PBS were injected intraperitoneally twice a week (day 0 and day 7) when the arthritis score reached 3 or more.

### 2.7. Flow Cytometry Analysis

The populations of Th17 and Treg cells were examined using flow cytometry. The The antibodies were phycoerythrin (PE)- conjugated rat anti-mouse CD4 (eBioscience, San Diego, CA, USA), fluorescein othiocyanate (FITC)- conjugated rat anti-mouseIL-17A (eBioscience), and FITC-conjugated rat anti-mouse CD25 (eBioscience). Analyses were conducted using CellQuest software (BD).

### 2.8. Histology and Analysis

To obtain cartilage and paw samples, mice were euthanized with CO_2_ gas and tissues were obtained by dissection. Limbs and Paw were fixed overnight in 4% paraformaldehyde and decalcified. Cartilage and paw were embedded in optimal cutting temperature (OCT) compound and cryosectioned at 10 µm. To analyze of inflammation, section was stained with Hematoxylin and Eosin (H&E) Staining. To confirm cartilage destruction of CIA model, the specimen was stained using safranin O/Fast green (Science cell) or staining following manufacturer’s instructions. Cartilage degradations were measured by cartilage degradation score using following scale: from 0 to 3 was defined as either no loss of proteoglycans or complete loss of staining for proteoglycans [18]. Pathologic changes are also scored by the degree of inflammation in cartilage and bone destruction according to the previous report [20] using the following scale: 0 = normal synovium, 1 = synovial membrane hypertrophy and cell infiltrates, 2 = pannus and cartilage erosion, 3 = major erosion of cartilage and subchondral bone, and 4 = loss of joint integrity and ankylosis.

### 2.9. Statistical Analysis

All data are presented as mean ± SD. Statistical analyses were conducted by Student’s *t*-test for comparisons of two groups, and ANOVA with Bonferroni’s multiple comparison test using SPSS v13.0. Data with *p* < 0.05 were considered statistically significant.

## 3. Results

### 3.1. Targeted Knock-In of IL-10 in AMMs

We used TALENs to generate a stable stem cells expressing IL-10. The targeting donor plasmid contained phosphoglycerate kinase promoter-driven IL-10, elongation factor-1-alpha, and promoter-driven green fluorescent protein (GFP)–T2A–puromycin (Appendix A). AMMs were transfected with a pair of TALENs and donor plasmids. Transfected cells were selected by culturing with puromycin (44.3%). In addition, 99.1% of GFP-positive cells were sorted using fluorescence-activated cell sorting (Appendix A). The correct insertion of the donor plasmid was confirmed by the detection of the five junction fragment (960 bp) amplification (Appendix A). Quantitative reverse transcription-polymerase chain reaction results also showed that IL-10 levels were significantly increased in AMM/I compared to those in untransfected AMMs (Appendix A). The IL-10 konck-in AMM cell line (AMM/I) was used in this study.

### 3.2. Characteristics of AMM/I

Cultured AMM/I displayed spindle fibroblast-like morphology similar to that of AMMs. To investigate the characteristics of AMM/I, we performed a flow cytometry analysis. AMM/I maintained the original characteristics of MSCs, expressing high levels of MSCs markers (CD29, CD73, and CD90) and very low levels of hematopoietic cell markers (CD14 and CD45) (Appendix A).

### 3.3. In Vitro Immunomodulatory Potential of AMM/I

To evaluate the anti-proliferation property of AMM/I, cell proliferation assay was performed using CD3+ splenocyte. Result showed that culture medium (CM) derived from AMM/I significantly inhibited the CD3+ T cells compare to the CM of AMM (Appendix A).

Next, to better investigate the in vitro immunomodulatory effects of AMM/I on T cells, AMM/I and AMMs were treated with or without TNF-α and co-cultured with splenocytes. Supernatants from co-cultures were used to determine the cytokine levels after two days. Enzyme-linked immunosorbent assay results showed that the co-culture with AMM/I significantly increased the IL-4 and decreased the IL-17A levels compared to those in the co-culture with AMMs (Figure 1).

### 3.4. Anti-Arthritogenic Property of AMM/I in a Collagen-Induced Arthritis (CIA) Mouse Model

To evaluate the anti-arthritogenic capacity of AMM/I in the articular cartilage in vivo, we induced arthritis in the paws of CIA mice. After induction of arthritis, AMM/I, AMMs, and phosphate-buffered saline (PBS) were intraperitoneally injected twice a week. (Figure 2A). Joint tissues were harvested 15 d after cell injection. Interestingly, the arthritis clinical score was significantly lower 15 d after injection of AMM/I compared with that resulting from a PBS injection or that in the AMM group (Figure 2B,C).

Next, to elucidate the mechanisms responsible for the anti-arthritogenic capacity of AMM/I, we analyzed the T cells. AMM/I injection increased the population of Tregs compared to PBS or AMM injection (Figure 3A,B). However, the population of Th17 cells significantly decreased in the AMM/I-treated group (Figure 3A,B). In addition, we measured the concentration of IL-17A in the serum of CIA mice following cell injection. AMM/I injection significantly decreased the levels of IL-17A compared to PBS or AMM injection (Figure 3C).

### 3.5. Histological Analysis of the Joints of CIA Mice after AMM/I Injection

To evaluate the potential for cartilage protection in vivo, joint tissues were stained with safranin O/fast green after cell injection. The AMM/I-treated group showed increased proteoglycan expression in the articular cartilage compared to the AMM- or PBS-treated control groups, indicating the protective property of AMM/I against cartilage damage (Figure 4A,B).

To examine the inflammatory response after cell injection, we performed hematoxylin and eosin staining. Histological analysis showed that AMM/I-injected articular tissue had significantly lower inflammatory cell infiltration than AMM- or PBS-treated articular tissues (Figure 4C,D).

### 3.6. Gene Expression Analysis of the Joints of CIA Mice after AMM/I Injection

To further elucidate the therapeutic mechanisms of AMM/I, we determined the expression levels of pro-inflammatory factors in the joint tissues of CIA mice after cell injection. Interestingly, pro-inflammatory factors, such as IL-1β, IL-6, monocyte chemoattractant protein (MCP)-1, and TNF-α, were significantly decreased in AMM/I-treated articular tissues (Figure 5).

## 4. Discussion

Although MSCs have attracted attention as novel therapeutic candidates for cartilage regeneration or protection, their therapeutic potential must be enhanced. In this study, we hypothesized that IL-10 overexpression may promote the therapeutic potential of MSCs in damaged cartilage. We demonstrated the anti-arthritogenic properties of IL-10-edited human AMMs in experimental arthritis.

Recently, MSCs were reported about their therapeutic role in the treatment of rheumatic diseases [21]. A few studies have reported the therapeutic efficacy of MSCs in the treatment of knee RA [22,23], while others have demonstrated that MSCs have little or no therapeutic effect in CIA mice [13,24,25], indicating that the immunosuppressive characteristics of MSCs are not sufficient to reverse the arthritic condition. High levels of IL-6 in the presence of TNF-α may impact the beneficial effects of MSCs in CIA [25]. Thus, the therapeutic effects of MSCs need to be enhanced for the treatment of arthritis, and we aimed to establish safe and effective methods to enhance the therapeutic properties of MSCs. Human AMMs mitigate RA progression [26]. However, AMMs do not show statistical significance in cartilage repair because of their low therapeutic potential [27]. Thus, we attempted to generate more potent and safer stem cell lines using gene editing for cartilage protection or repair.

To enhance the therapeutic potential of stem cells without abnormal mutation risk, we developed a gene knock-in system using AMMs and gene-editing method [15]. In particular, human AMMs were widely studied for the allogeneic stem cell therapy due to the property of immune privilege and high differentiation potential [28]. In addition, AMMs are suitable for gene editing because of their highly efficient gene insertion and cell proliferation potential. Thus, we used AMMs in this study.

IL-10 is a representative anti-inflammatory cytokine that plays a protective role in inflammatory arthritis and inhibits the progression of CIA [29]. The treatment with murine IL-10-expressing MSCs profoundly decreases the production of inflammatory cytokine, IL-6, while increasing the level of anti-inflammatory cytokine, IL-4 in the splenic cells of CIA mice [13]. In agreement with these data, highly increased levels of IL-4 or decreased levels of inflammatory cytokine, Th-17, were detected in the co-culture with AMM/I and splenocytes in an inflammatory environment. Thus, we hypothesized that stable overexpression of IL-10 may enhance the anti-arthritogenic potential of AMMs. Interestingly, we observed that gene-edited AMM/I inhibited the severity of arthritis in collagen-induced mice, indicating that AMM/I may be alternative therapeutic agents for treating RA. These data are similar to those of previous reports that IL-10 is involved in the suppressive effects on synoviocytes and IL-10-secreting bone marrow-derived MSCs attenuate CIA in mice [13,23]. Additionally, our results are similar to those of a previous report that MSCs show low or no therapeutic benefit in CIA mice [13,24], suggesting that the immunosuppressive characteristics of MSCs are not sufficient to reverse the arthritic condition.

T cells play an important role in the regulation of inflammatory responses in RA. Th1 and Th17 cell-mediated responses are associated with the pathogenesis of CIA [5]. In addition, Tregs play critical roles in the prevention of autoimmunity and modulation of CIA [30]. IL-10 inhibits T cell proliferation. Immunomodulatory factors suppress inflammation by reducing the proliferation of Th17 cells and inducing Tregs [31]. Consistent with previous reports, we found that AMM/I administration downregulated Th17 cells, suggesting arthritis suppression by IL-10 derived from AMM/I. In contrast, we found that AMM/I induced Tregs in CIA mice. These results indicate that the anti-inflammatory function of IL-10 may involve the reciprocal effect of Th17/Treg cell imbalance in CIA.

One of the most critical points for the therapeutic effect of MSCs is the change in the disease-specific inflammatory milieu. High concentrations of TNF-α, IL-1 β, IL-6, and MCP-1 have been detected in the synovial fluid of patients with RA [9,32]. Transplantation of human umbilical cord blood MSCs reduces the NLR family pyrin domain containing 3 inflammasome-mediated IL-1β production in macrophages of CIA mice by enhancing cyclooxygenase-2 signaling in response to IL-1β [9]. In addition, systemic infusion of human umbilical cord MSCs reduces the levels of TNF-α, IL-6, and MCP-1 in CIA mice [23]. In line with these reports, we found that AMM/I injection downregulated the levels of the proinflammatory cytokines, TNF-α, IL-1β, IL-6, and MCP-1, in CIA mice. Reduction in the levels of these proinflammatory cytokines can explain the inhibition of inflammation in the synovium of AMM/I-treated mice.

In conclusion, AMM/I exerted profound anti-arthritogenic effects by suppressing T cell activation, inhibiting inflammatory responses, and inducing Treg generation. Most importantly, transplantation of AMM/I significantly ameliorated CIA in mice. These data indicate that human AMM/I may potentially be used as therapeutic agents for RA.

## Figures and Tables

**Figure 1 ijms-23-07913-f001:**
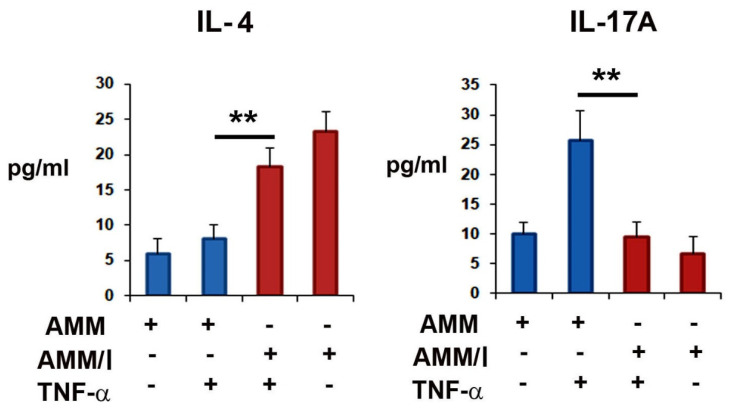
Immunomodulatory potential of interleukin (IL)-10-overexpressing amniotic mesenchymal stem cells (AMM/I) in vitro. AMMs and AMM/Id were treated or not with tumor necrosis factor (TNF)-α, and then they were co-cultured with splenocytes. The supernatants were collected and the concentrations of IL-10 and IL-17A were measured via enzyme-linked immunosorbent assay (ELISA) (n = 5 each; ** *p* < 0.01).

**Figure 2 ijms-23-07913-f002:**
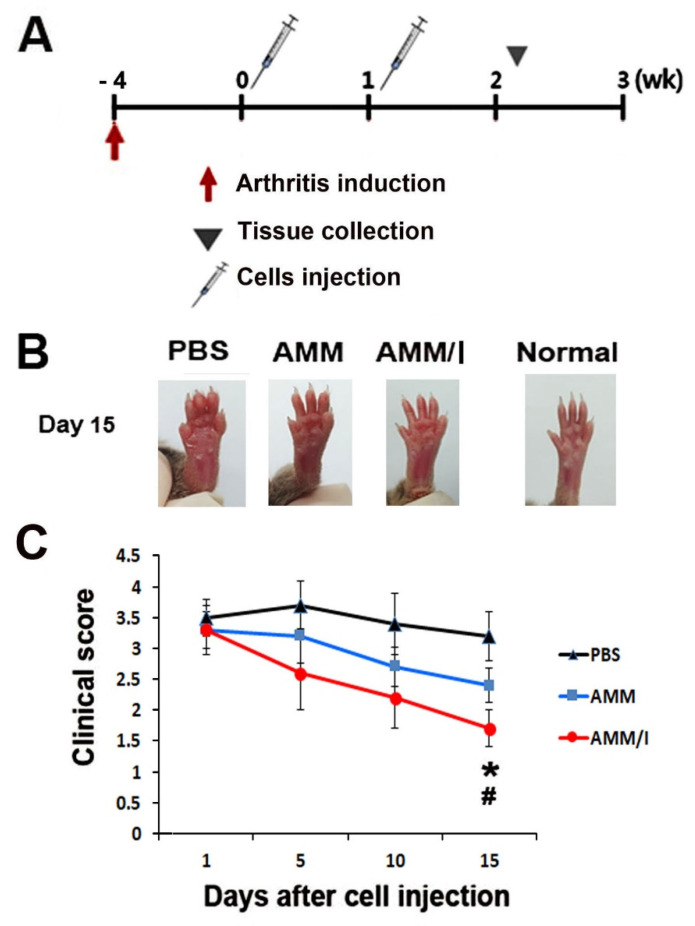
AMM/I transplantation leads to the prevention of disease progression. (**A**) Schematic representation of the procedures for the induction of arthritis, cell injection, and collection of specimens. (**B**) Representative pictures of mice paws after cell injection. (**C**) Quantification of arthritis scores. The arthritis scores were measured using severe swelling paws. # *p* < 0.01 AMM/I vs. PBS,* *p* < 0.05 AMM vs. AMM/I, n = 5 per group.

**Figure 3 ijms-23-07913-f003:**
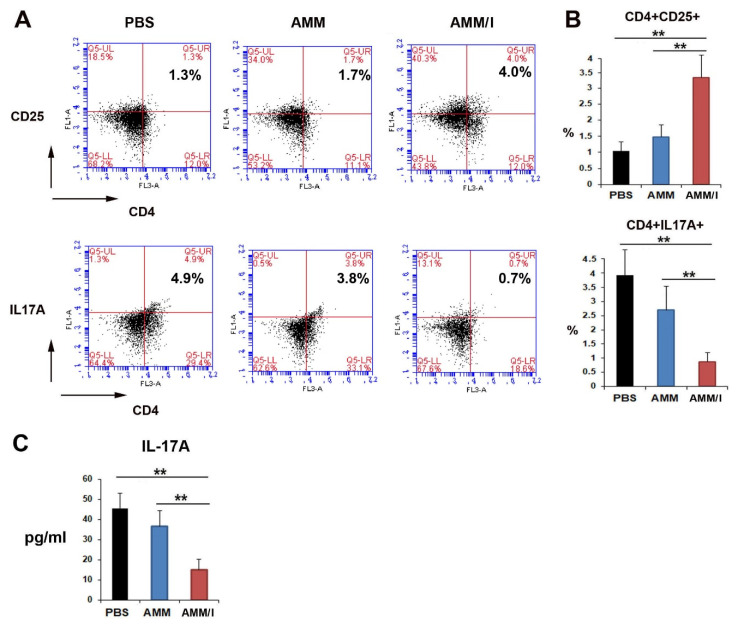
AMM/I transplantation influences the populations of regulatory T cells (Tregs) and helper T (Th)-17 cells in collagen-induced arthritis (CIA) mice. (**A**) Representative figure of flow cytometry data for the identification of Tregs and Th17 cells. (**B**) Quantitative data for Tregs and Th17 cells were measured using CIA mouse blood samples two weeks after the injection of cells (n = 5 each; ** *p* < 0.01). (**C**) The concentration of IL-17A in CIA mice two weeks after cells injection (n = 5 each; ** *p* < 0.01).

**Figure 4 ijms-23-07913-f004:**
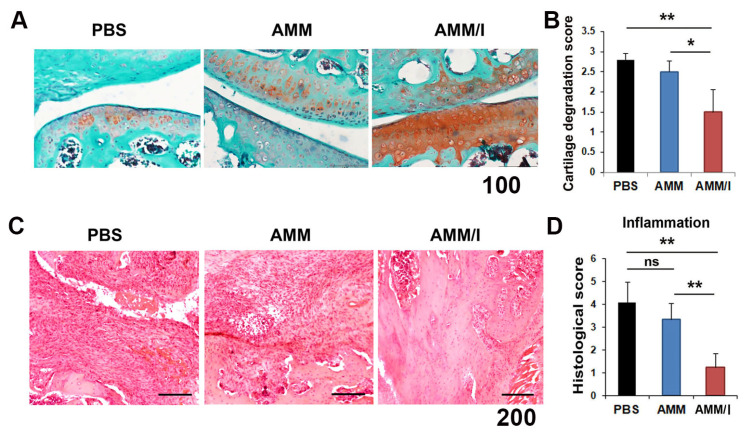
Histological staining of the joints of CIA mice after the injection of cells. (**A**) Proteoglycan expression was determined using safranin O/fast green staining of the joints of CIA mice after cell injection. Bars = 200 μm. (**B**) Quantification of the cartilage degradation score. Loss of proteoglycans was determined by staining the proteoglycans (n = 5 each; * *p* < 0.05, ** *p* < 0.01). (**C**) Representative pictures of hematoxylin and eosin (H&E)-stained sections of joint tissues. Bars = 200 μm. (**D**) Quantification of the inflammatory pathological score. Mononuclear cell infiltration and inflammatory pathological scores were measured after cell transplantation. AMM/I-injected tissues showed low mononuclear cell infiltration and a normal cartilage surface morphology (n = 5 each; ** *p* < 0.01). ns, not significant.

**Figure 5 ijms-23-07913-f005:**
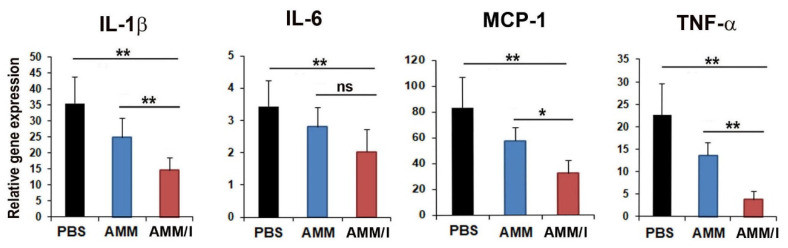
AMM/I transplantation suppresses the inflammation of mouse joints. Quantitative reverse transcription-polymerase chain reaction (qRT-PCR) analyses of joint tissues injected with phosphate-buffered saline (PBS), AMMs, and AMM/I. AMM/I transplantation resulted in decreased expression levels of representative pro-inflammatory factors and increased expression levels of anti-inflammatory factors. (n = 5 each; * *p* < 0.05, ** *p* < 0.01). ns, not significant.

## Data Availability

Not applicable.

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
