# Peer review of "Anti-Arthritogenic Property of Interleukin 10-Expressing Human Amniotic MSCs Generated by Gene Editing in Collagen-Induced Arthritis"

_ijms, 2022, doi:10.3390/ijms23147913_

Round 1

Reviewer 1 Report

In this study, Dong-Sik and colleagues explored the potential role of L-10-overexpression in rheumatoid arthritis both in vitro by co-cultured experiments and in vivo on collagen-induced arthritis (CIA) mice by using human amniotic mesenchymal stem cells (AMMs) generated via gene editing. They found that AMMs overspressing IL-10 exerted profound anti-arthritogenic effects by suppressing T cell activation, inhibiting inflammatory responses, and inducing Treg generation. The authors suggest that these data indicate that human AMM/Is may potentially be used as therapeutic agents for RA.

Major:

-          - Phenotypic characterization of MSCs should be showed. To confirm human MSC phenotype as showed here (Cipriani P, Clin Exp Immunol. 2013 doi: 10.1111/cei.12111.), other markers including CD34 and CD105 should be used. The authors should explain why they decide to use Human amniotic mesenchymal stem cells (MSC) instead of MSC derived from other tissues (for example adipose tissue) of RA patients.

-        -   In introduction and discussion section the authors should mention the role of MSC in others autoimmune diseases. Paper from Cipriani P and colleagues - > PMID: 25680301 DOI: 10.1016/j.jcyt.2014.12.006, should be referenced.

--  The author showed the effect of IL-10 overexpression on cartilage degradation in mice; however, in order to explore the mechanisms by which IL-10 could prevent cartilage destruction in RA, it should be interesting to explore in vitro the effect of MSCs overespressing IL-10 on primary chondrocytes from RA patients (pro-inflammatory cytokines levels, matrix metalloproteinases expression, signaling pathway analalysis).

-         -  To evaluate the immunosuppressive activity of AMM/Is, the effect on proliferation (for example by Ki-67 assay) and on apoptosis of T cells should be assessed.

-         -  The analysis of Treg cells is not clear. In material and methods section, the authors say that they used FITC-conjugated rat anti-mouse Foxp3 (eBioscience), but in Figure 3 they didn’t show this marker and, in the graph (Figure 3B), Treg cells are expressed as CD4+ CD25+ cells (not CD4+, CD25+, FoxP3+). In addition, it would have been interesting to evaluate the frequencies of nTreg, eTreg, and non-Treg fractions, as showed here Scrivo R, et al. PLoS One. 2017.  Furthermore, the protocol used for Treg and Th17 evaluation should be better explained.

Minor

-    -       “we analyzed the T cells in the serum”, This sentence is not correct

-      -    On Figure 3, the authors miss the legend of C section.

Author Response

Reviewer’s comment

In this study, Dong-Sik and colleagues explored the potential role of L-10-overexpression in rheumatoid arthritis both in vitro by co-cultured experiments and in vivo on collagen-induced arthritis (CIA) mice by using human amniotic mesenchymal stem cells (AMMs) generated via gene editing. They found that AMMs overspressing IL-10 exerted profound anti-arthritogenic effects by suppressing T cell activation, inhibiting inflammatory responses, and inducing Treg generation. The authors suggest that these data indicate that human AMM/Is may potentially be used as therapeutic agents for RA.

Major:

-          - Phenotypic characterization of MSCs should be showed. To confirm human MSC phenotype as showed here (Cipriani P, Clin Exp Immunol. 2013 doi: 10.1111/cei.12111.), other markers including CD34 and CD105 should be used. 

Response

We added the AMM phenotypes including CD34 and CD105 in Supple Fig.2.

The authors should explain why they decide to use Human amniotic mesenchymal stem cells (MSC) instead of MSC derived from other tissues (for example adipose tissue) of RA patients.

 Response

AMMs were widely studied for the allogeneic stem cell therapy due to the property of immune privilege and high differentiation potential (Magatti et al., 2008). In addition, AMMs are suitable for gene editing because of their highly efficient gene insertion and cell proliferation potential.                      Thus, we decided to use AMMs in this study. We added this in the Discussion section.

-       In introduction and discussion section the authors should mention the role of MSC in others autoimmune diseases. Paper from Cipriani P and colleagues - > PMID: 25680301 DOI: 10.1016/j.jcyt.2014.12.006, should be referenced.

 Response 

We added the reference of paper from Cipriani P  in the discussion section.

-  The author showed the effect of IL10 overexpression on cartilage degradation in mice; however, in order to explore the mechanisms by which IL-10 could prevent cartilage destruction in RA, it should be interesting to explore in vitro the effect of MSCs overespressing IL-10 on primary chondrocytes from RA patients (pro-inflammatory cytokines levels, matrix metalloproteinases expression, signaling pathway analalysis).

 Response          

To performe the experiment the effect of AMM/I on primary chondrocytes from RA patients, we used 

conditioned medium (CM) of AMM/I and AMM and perforemd qRT- PCR.

But we could not find any differences of pro-inammation and MMP expression in the chondrocytes co-cultured with CM of AMM/I and AMM.

                         Reviewer's Figure 

To demonstrate the therapeutic mechanism of AMM/I in vitro, we investigated the the in vitro immunomodulatory effects of AMM/Is on T cells (Figure 1). 

Results showed that highly increased levels of inflammatory cytokine, IL4 or decreased levels of inflammatory cytokine Th17, were detected in the co-culture with AMM/I and splenocytes in an inflammation environment. In addition, to further elucidate the therapeutic mechanisms of AMM/Is, we determined the expression detected in the coculture with AMM/I and splenocytes in an inflammatory environment. In addition, to 

further elucidate the therapeutic mechanisms of AMM/Is, we determined the expression levels of proinflammatory factors in joint tissues of CIA mice after cell injection (Figure 5).                                                                                                                

We speculate that these data can provide the therapeutic mechanism of AMM/I for the cartilage destruction in RA.

-   To evaluate the immunosuppressive activity of AMM/Is, the effect on proliferation (for example by Ki-67 assay) and on apoptosis of T cells should be assessed.

 Response

We tried to do proliferation assay using ki-67, but we failed. Thus, we performed T cell proliferation assay by CCK assay. Results were added and showed in the Supplementary Figure 3.

-     The analysis of Treg cells is not clear. In material and methods section, the authors say that they used FITC-conjugated rat anti-mouse Foxp3 (eBioscience), but in Figure 3 they didn’t show this marker and, in the graph (Figure 3B), Treg cells are expressed as CD4+ CD25+ cells (not CD4+, CD25+, FoxP3+). In addition, it would have been interesting to evaluate the frequencies of nTreg, eTreg, and non-Treg fractions, as showed here Scrivo R, et al. PLoS One. 2017.  Furthermore, the protocol used for Treg and Th17 evaluation should be better explained.

Response

There was typing error. It was FITC-conjugated rat anti-mouse CD25 (eBioscience). We revised this in the method section.

Minor

-      “we analyzed the T cells in the serum”, This sentence is not correct

 Response

We revised this sentence.

-        On Figure 3, the authors miss the legend of C section.

 Response 

We added the legend of C section.

Reviewer 2 Report

The authors' aim in this article is to prove the therapeutic proprieties of IL-10 overexpressed by human amniotic mesenchymal stem cells (AMM/I), created by gene editing. The authors show the results obtained in a collagen induced arthritis mouse model and results obtained   by in vitro study of immunomodulation proprieties of AMM/I. The research design is appropriate, all materials and methods are sufficiently described, the conclusions are supported by results.  The results are clearly presented but there are few errors. Test editing is also required for one supplementary figure legend. A more extensive  introduction could better could contestualized the results

1) Introduction: Characterize better and more extensively the Rheumatoid arthritis

2) the author’s name, the abstract, the quantitative reverse transcription PCR (qRT-PCR), Fig. 5 and results 3.1. (Targeted knock-in of IL-10 in AMMs) are written in a different font, please correct it

3) It is necessary to correct supplement legend fig. n1: donor vector carries IL-10 and not TGF-B.

4) In material and methods in Targeted knock -in of IL-10 in AMMs are not indicated negative and positive controlls

5) Add legend to original image (supplement figure 1A) and indicate the samples into gel.

6) images of electrophoresis or blot, it is not clear. Please add legend and indicate the samples in figure 1B (image of gel and image of blot) in supplement data

7) please, correct fig 1B and 1C…. 1B is 1C and 1B is 1C?

8) Fig 3 A? dot blot is not correct if AMM/I induces up regulation of Treg CD25/CD4, as reported in Fig 3B

9) please, if you have, show characterization of Culture AMM/I cells by flow cytometry (data not shown). it is important to verify no alterations in AMM/I cells in respect to AMM

Author Response

Reviewer’s comment                    

The authors' aim in this article is to prove the therapeutic proprieties of IL-10 overexpressed by human amniotic mesenchymal stem cells (AMM/I), created by gene editing. The authors show the results obtained in a collagen induced arthritis mouse model and results obtained   by in vitro study of immunomodulation proprieties of AMM/I. The research design is appropriate, all materials and methods are sufficiently described, the conclusions are supported by results.  The results are clearly presented but there are few errors. Test editing is also required for one supplementary figure legend. A more extensive  introduction could better could contestualized the results

1) Introduction: Characterize better and more extensively the Rheumatoid arthritis

Response

We revised.

2) the author’s name, the abstract, the quantitative reverse transcription PCR (qRT-PCR), Fig. 5 and results 3.1. (Targeted knock-in of IL-10 in AMMs) are written in a different font, please correct it

Response

We revised.

3) It is necessary to correct supplement legend fig. n1: donor vector carries IL-10 and not TGF-B.

Response

We revised in the supplement legend figure 1.

4) In material and methods in Targeted knock -in of IL-10 in AMMs are not indicated negative and positive controlls

Response

We revised.

5) Add legend to original image (supplement figure 1A) and indicate the samples into gel.

Response

We revised.

6) images of electrophoresis or blot, it is not clear. Please add legend and indicate the samples in figure 1B (image of gel and image of blot) in supplement data

Response

We revied this

7) please, correct fig 1B and 1C…. 1B is 1C and 1B is 1C?

Response

We revied this

8) Fig 3 A? dot blot is not correct if AMM/I induces up regulation of Treg CD25/CD4, as reported in Fig 3B

Response

We don’t understand this. There no dot blot in Fig 3.

9) please, if you have, show characterization of Culture AMM/I cells by flow cytometry (data not shown). it is important to verify no alterations in AMM/I cells in respect to AMM

Response

We added flow cytometry data of AMM/I in Supple Figure 2.